# Virtual Screening-Based Peptides Targeting Spike Protein to Inhibit Porcine Epidemic Diarrhea Virus (PEDV) Infection

**DOI:** 10.3390/v15020381

**Published:** 2023-01-28

**Authors:** Qian Xu, Fangyu Wang, Wenqiang Jiao, Mengting Zhang, Guangxu Xing, Hua Feng, Xuefeng Sun, Man Hu, Gaiping Zhang

**Affiliations:** 1Department of Preventive Veterinary Medicine, College of Veterinary Medicine, Northwest A&F University, Yang ling, Xianyang 712100, China; 2Key Laboratory for Animal Immunology, Henan Academy of Agricultural Sciences, 116# Huayuan Road, Zhengzhou 450002, China; 3Longhu Modern Immunology Laboratory, Zhengzhou 450046, China; 4School of Advanced Agricultural Sciences, Peking University, Beijing 100871, China; 5Jiangsu Co-Innovation Center for the Prevention and Control of Important Animal Infectious Diseases and Zoonoses, Yangzhou University, Yangzhou 225009, China

**Keywords:** porcine epidemic diarrhea virus, S1 C-terminal domain (CTD) protein, virtual screening, SPR, antiviral peptides, qRT-PCR, indirect immunofluorescence

## Abstract

Due to the rapid mutation of porcine epidemic diarrhea virus (PEDV), existing vaccines cannot provide sufficient immune protection for pigs. Therefore, it is urgent to design the affinity peptides for the prevention and control of this disease. In this study, we made use of a molecular docking technology for virtual screening of affinity peptides that specifically recognized the PEDV S1 C-terminal domain (CTD) protein for the first time. Experimentally, the affinity, cross-reactivity and sensitivity of the peptides were identified by an enzyme-linked immunosorbent assay (ELISA) and a surface plasmon resonance (SPR) test, separately. Subsequently, Cell Counting Kit-8 (CCK-8), quantitative real-time PCR (qRT-PCR), Western blot and indirect immunofluorescence were used to further study the antiviral effect of different concentrations of peptide 110766 in PEDV. Our results showed that the P/N value of peptide 110766 at 450 nm reached 167, with a K_D_ value of 216 nM. The cytotoxic test indicated that peptide 110766 was not toxic to vero cells. Results of the absolute quantitative PCR revealed that different concentrations (3.125 μM, 6.25 μM, 12.5 μM, 25 μM, 50 μM, 100 μM, 200 μM) of peptide 110766 could significantly reduce the viral load of PEDV compared with the virus group (*p* < 0.0001). Similarly, results of Western blot and indirect immunofluorescence also suggested that the antiviral effect of peptide 110766 at 3.125 is still significant. Based on the above research, high-affinity peptide 110766 binding to the PEDV S1-CTD protein was attained by a molecular docking technology. Therefore, designing, screening, and identifying affinity peptides can provide a new method for the development of antiviral drugs for PEDV.

## 1. Introduction

Porcine epidemic diarrhea (PED) was first identified in Europe in the early 1970s, and the virus was first isolated in Belgium in 1978 [1]. It was reported that PEDV had broken out in many swine-producing regions including China [2,3], the USA [4], Canada [5]; and Europe [6,7], it causes severe enteric disease that inflicts huge economic damage on the swine industry worldwide. It encodes four main structural proteins: spike (S), nucleocapsid(N), membrane (M) and envelope (E). Among these, S protein is processed into S1 and S2 subunits by trypsin-like host cell proteases, which play an important role in the mediator of viral attachment to host cells [8]. S1-CTD region in PEDV S protein is one of the key targets for antiviral drug developments as to PEDV. However, the existing PED vaccines could not provide appropriate protection against the epidemic PEDV infection. Considering this factor, studies against new PEDV strains are necessary to prevent and control emerging or re-emerging infectious diseases.

For most coronaviruses, the N-terminal domain (NTD) of the S1 subunit attaches to cellular carbohydrates and the C-terminal domain binds to a cellular protein receptor [9,10,11,12,13]. The receptor-binding domain (RBD) in the S1 subunit of SARS-CoV-2 is responsible for viral receptor engagement through with host cell receptor(s), but related studies of coronavirus infection have reported that synthetic peptides targeting the RBD domain from SARS-CoV-2 could block viral entry into cells [14]. Therefore, RBD is an essential target for the development of viral attachment inhibitors, even including neutralizing antibodies (nAbs). The S1-CTD domain in the S1 subunit of PEDV spike (S) protein can stimulate the animal body to produce neutralizing antibodies, which is also a relatively conservative target for the development of an antiviral peptide.

The structure of peptides is relatively simple and the molecular weight is small; therefore, they are easy to synthesize and modify [15], have higher cell membrane penetration, no cytotoxicity and less immunogenicity [16]. Common methods for screening peptides are phage display technology, mRNA display technology, combinatorial chemistry technology, computer-based virtual screening technology and so on [17]. However, these methods are highly dependent on high-throughput experimental screening, which readily increase workload. whereas the use of structure-based molecular docking technology could overcome these problems. Molecular docking is one of the key techniques of computational virtual screening, which tries to predict the binding mode and affinity of a ligand to the active site of a protein [18]. It possesses many advantages, such as simple and fast operation, which reduces the workload of peptide screening, shortens the development cycle, and improves the screening success rate [19,20].

In this study, we, for the first time, designed and synthesized the peptide targeting porcine epidemic diarrhea virus (PEDV)S1-CTD region by making use of computer virtual screening technology to screen the optimal peptide via an Enzyme Linked Immunosorbent Assay (ELISA) and surface plasmon resonance (SPR) experiment. Furthermore, a cytotoxicity assay, fluorescence quantitative PCR, Western blotting and indirect immunofluorescence techniques were used to evaluate antiviral activity for these peptides in a PEDV epidemic strains infection. Five peptides (115042, 111740, 110766, 110616 and 110490) were able to inhibit PEDV infection, with 110766 being the most potent.

## 2. Materials and Methods

### 2.1. Cells and Viruses

African green monkey kidney cells (Vero E6) were cultured in Dulbecco’s modified Eagle’s medium (DMEM, Solarbio) supplemented with 10% fetal bovine serum (FBS, Gibco) at 37 °C in a 5% CO_2_ incubator. PEDV strain CH-hubei-2016 (GenBank Accession No. KY928065.1) was kept in our laboratory.

### 2.2. Biological Materials

The peptides were screened by molecular docking in our laboratory by SYBYL-X 2.0 software, synthesized, and purified to at least 90% by GL Biochem (Shanghai, China). The porcine epidemic diarrhea virus S protein (PEDV-S1), PEDV S1 protein monoclonal antibody, anti-his tag mouse monoclonal antibody were made and PEDV N protein monoclonal was conserve in our laboratory, porcine circovirus 2 Cap protein (PCV2-Cap), porcine reproductive and respiratory syndrome virus GP5 protein (PRRSV-GP5) were generated in our laboratory. TransZol Up and PerfectStartIIProbe qPCR SuperMix UDG were purchased from TransGen Biotech (Beijing China). The goat anti-mouse IgG conjugated to horseradish peroxidase (IgG-HRP) was purchased from Abbkine (Wuhan, China) antibody goat anti-mouse IgG(H+L) highly cross-adsorbed secondary antibody, Alexa Fluor Plus 488, was purchased by Thermo Fisher Scientific (Invitrogen, Waltham, MA, USA). The cell-counting kit-8 (CCK-8), 2-(4-Amidinophenyl)-6-indolecarbamidine dihydrochloride (DAPI) and RIPA lysis buffer were purchased from Beyotime (Shanghai, China).

### 2.3. Analysis of the S1-CTD Amino Acid Sequence of PEDV

To evaluate the conservation of PEDV S1-CTD domain, 10 PEDV variant strains amino acid sequences were aligned and analyzed by using the MegAlign tool in Lasergene DNASTAR™ 5.06 software (named HB S1-CTD, GDS28 S1-CTD, CH JX S1-CTD, ZMDZY S1-CTD, AH2012 S1-CTD, AH2012-12 S1-CTD, CHSD2014 S1-CTD, LC S1-CTD, AJ1102 S1-CTD, CH SGZG S1-CTD, respectively) (Table 1).

### 2.4. Molecular Docking Virtual Screening

Methods of designing and screening of all the peptides were described in previous articles [21,22]. (1) Preparation of protein crystal structure. The 3D structure of PEDV S protein (PDB ID:6U7K) [23] was derived from the protein database; (2) Virtual peptide library design. All peptides were designed as a series of linear peptides about 2 to 9 amino acids [24]; (3) Molecular docking. By using the Surflex-Dock program in SYBYL software, PEDV S was prepared by Preparing Protein Structure embedded in the SYBYL; and (4) Evaluation of results. The affinity between peptides and PEDV S1-CTD protein were evaluated through Cscore values calculated by consensus score function and the interactions between peptides and PEDV S1-CTD protein were analyzed by View/Surfaces and Ribbons/Create/MOLCAD module in SYBYL-X 2.0 software [24]. Therefore, based on the criterion of virtual screening (CScore ≥ 5), we obtained 53 peptide sequences.

### 2.5. ELISA Assay for Affinity and Specificity

The purified PEDV S1 protein, PRRSV-GP5 protein, PCV2-Cap protein, and the BSA powder were separately diluted with carbonate buffer saline (CBS) to 10 μg/mL. The ELISA plates were coated with the PEDV S1 protein solution, PRRSV-GP5 protein solution, PCV2-Cap protein solution, and the BSA solution 100 μL per well, named S1-plate GP5-plate, Cap-plate or BSA-plate, and incubated at 37 °C for 2 h. The plates were washed with PBST (phosphate buffer saline, PBS containing 0.5% Tween20) five times, and blocked with the blocking buffer, the PBST solution containing 5% skimmed milk powder, 100 μL per well and incubated at 37 °C for 1 h. After washing, the plates were stored at 4 °C.

The his-labeled peptides, diluted with PBS to 1 μg/mL, were respectively added and incubated for 1 h at 37 °C. Then, the plates were washed as before, and the anti-his tag mouse monoclonal antibody (1:1000) was added at 100 μL per well and incubated for 30 min at 37 °C. BSA were used as negative controls, and an anti-PEDV S1 monoclonal antibody was used as a positive control. PBS without peptides were used as blanks. After washing, the 3,3,5,5-tetramethylbenzidine (TMB) substrate was used as a color agent for 10 min. The reaction was stopped with 2 M H_2_SO_4_ solution. The OD value of each well was measured at 450 nm using a microplate reader. Meanwhile, the cross-reactivity of affinity peptides was evaluated using proteins outside the circovirusgenus, namely PRRSV-GP5 protein, PCV2-Cap protein and BSA. Their coating concentration was 10 µg/mL.

When the ratio between P/N [(OD_sample_ − OD_blank_)/(OD_negative_ − OD_blank_)] is more than 2.1, the P/N value of the candidate peptides were determined according to the OD value, and the peptides exhibiting high affinity were chosen for posterior experiments.

### 2.6. Kinetic Dissociation Measurements

All the experimental procedures were set according to the Biacore X100 Manual. The optimal pH of sodium acetate for coupling PEDV S1 protein was determined by pre-experiment at the first, then PEDV S1 protein was covalent coupled to a carboxyl Au colloidal nanoparticles (C-AuNPs) chip (CM5 chip) by the EDC/NHS method at the optimal pH. The running buffer (HBS-EP, pH7.4, was filtered with a 0.22 μm filter) was then flowed through the CM5 chip. The double ratio dilution method by HBS-EP was applied to different peptides, which were injected from low concentrations to high concentrations to detect the resonance signal changes. In each cycle, the peptide solution was flowed through the chip for 120 s at a constant flow rate of 30 μL/min, and HBS-EP flowed via the chip for 120 s to dissociate the peptides from the PEDV S1 protein. 0.25% SDS was used to completely elute the peptides from the PEDV S1 protein [17]. Finally, the kinetic dissociation constant (K_D_) from peptides of the binding reactions were determined by Biacore X100 Evaluation Software (General Electric Company, Boston, MA, USA).

### 2.7. Cytotoxicity Assay

Vero cells were cultured in 96-well plates at a density of 1.0 × 10^4^ cells/well, followed by additional serial dilutions (0.78125–400 µM) of the peptide 110766. After incubation at 37 °C for 24 h, the cell supernatant was removed and the plate was washed three times with PBS, then 100 μL fresh medium containing 10 μL cell counting kit 8 (CCK-8) was added to each well and incubate with the cells at 37 °C for 1 h.

### 2.8. Absolute Quantification Real-Time PCR Assay

To evaluate the antiviral effect of peptide 110766 in PEDV-infected vero cells, PEDV at MOI 0.01 was first incubated with increasing concentrations of the peptides (3.125, 6.25, 12.5, 25, 50, 100 and 200 μM) for 1 h at 37 °C and then inoculated onto the cells. The inoculums were replaced with fresh DMEM containing 2% FBS and incubated for 12 h. Total RNAs were extracted from the cells in a 24-well plate by using TransZol Up according to the manufacturer’s instructions. Reverse transcription was performed as previously described [25] using 10 μL of the reaction mixture containing 4.5 μL of the total RNA extraction solution, 2 μL 5×PrimeScript RT Master Mix (TaKaRa, Japan), 3.5 μL RNase-Free Water after 20 min kept under 37 °C, 5 s kept under 85 °C, 16 °C storage. The resulting cDNA was finally stored at −20 °C. The cDNA of all samples was performed by the absolute quantification real-time PCR method, as previously described, namely, using 20 μL of the reaction mixture containing 2 μL cDNA, 0.4 μL upstream and downstream primers, respectively, 10 μL PerfectStart II Probe qPCR SuperMix UDG, 0.4 μL Probe, 6.4 μL nuclease-free water, 0.4 μL RoxReference Dye I(50×). The amplification was carried out as follows: 94 °C for 5 min, followed by 40 cycles of 94 °C for 5 s, 60 °C for 30 s (signal sampling).

### 2.9. Western Blot Assay

The mode of action of a mixture of peptide 110766 and PEDV on vero cells was the same as that in Section 2.8. Finally, the proteins were harvested through RIPA lysis buffer. Equal protein amounts were separated by sodium dodecyl sulfate polyacrylamide gel electrophoresis (SDS-PAGE), and after Western blot analysis, proteins were detected using specific antibodies against PEDV N protein and β-actin and the Enhanced Chemiluminescence (ECL) Detection System. PEDV N protein signal intensities were normalized to β-actin and quantified using Image J software.

### 2.10. Immunofluorescence Assay

The mode of action of a mixture of peptide 110766 and PEDV on vero cells was the same as that in Section 2.8. Vero cells cultured in 24-well plates were washed with PBST three times and fixed with 4% paraformaldehyde for 30 min at 4 °C, 0.02% triton-X 100 in PBS for 20 min at RT, and were then blocked with 5% nonfat-dried milk for 2 h at 37 °C and incubated with PEDV N protein monoclonal antibody (1:1000) for 1 h at 37 °C as well as the fluorescence-conjugated goat anti-mouse IgG (1:1000) for 1 h at 37 °C, and followed by the incubation with 2-(4-Amidinophenyl)-6-indolecarbamidine dihydrochloride (DAPI) for 15 min at RT. Fluorescence signals were captured with a fluorescence microscope.

### 2.11. Statistics

All experiments were used with three independent experiments, and the calculated results were presented as mean ±standard deviation (SD). Statistical analyses were performed by using one-way ANOVA. Graph Pad Prism8.0 was used to analyze the statistics in this study. The statistical significances were defined as *p* < 0.05 (*), and the higher significance was denoted by *p* < 0.01 (**), *p* < 0.001 (***), *p* < 0.0001 (****).

## 3. Results

### 3.1. Design and Synthesis of Peptides

PEDV S1-CTD domain is mainly localized in S1 protein C-terminal (residues 504–637) and different PEDV variant strains amino acid sequence homology in this region are higher than 96% (Figure 1A). Thus, the amino acid region at positions 505–642 (Figure 1B red part) of PEDV S1 protein C-terminal was selected as the docking pocket (Figure 2A), According to the results of preliminary screening of the experiments, five high-affinity candidate peptides were obtained and synthesized from 53 peptide sequences, among them sequences and CScore values are listed in Table 2. The 110766 peptide consists of two hydrophobic amino acids (Phe and Trp), three basic amino acids (two Lysines and Arg) and one uncharged amino acid (Ser). Its 2D structure is shown in Figure 2D. The multiple hydrogen bond (H-bonds, yellow dotted line) of the interaction between peptide 110766 and PEDV S1-CTD is shown in Figure 2B,C. The key amino acids at the binding site of PEDV S1-CTD protein were 550-Leu, 552-Tyr, 561-Val and 563-Lys.

### 3.2. Affinity and Specificity Analysis of Peptides by ELISA

According to the results ELISA assay, OD values at 450 nm of the 12 peptides, named 114180, 116265, 112542, 111692, 110766, 116306, 111885, 110616, 113596, 115042, 110490, 111740 were ranged from 0.2327 to 1.3193 and it of the monoclonal antibody to PEDV S1 protein as a positive control was 1.4493(Figure 3A). The five peptides with the highest OD values to PEDV S1 protein were 115042, 111740, 110766, 110616 and 110490 from high to low, indicating these five peptides had higher affinity. All these five peptides were specific with no reaction with BSA, and the P/N values were all greater than 67.882. Notably, the P/N value of 110766 was reached 167 (Figure 3B). The cross-reactivity of these five peptides to PRRSV-GP5 protein, porcine circovirus type II Capsid (PCV2-Cap) were shown in the (Figure 3C). Compared with the other four peptides, 110766 exhibited a lower cross-reactivity.

### 3.3. Measures for the K_D_ Values of Peptides by SPR

The optimal pH of sodium acetate was determined at the pH4.0 for coupling PEDV S1 protein by pre-experiment (Figure 4A) and then PEDV S1 protein was also covalent-coupled to carboxyl Au colloidal nanoparticles (C-AuNPs) chip (CM5 chip) (Figure 4B). According to the “1:1 binding” fitted curves of different concentrations of peptides (concentrations ranged from 0.78125 μM to 25 μM, respectively), as shown in (Figure 5A–E), the kinetic constants of the peptide-protein binding were calculated. The above five peptides showed different binding abilities to PEDV S1 protein with different K_D_ values (Table 3). Importantly, the K_D_ values of 110766 reached 216 nM. These data suggested that all the five peptides had high affinity to PEDV S1 protein, which were consistent with ELISA results.

### 3.4. The Inhibitory Effect of 110766 against HB-PEDV Strain

To fully understand the inhibitory effect of peptides (110766) against HB-PEDV strain, several experiments were performed including cytotoxicity test qRT-PCR, Western-Blot and IFA assay.

#### 3.4.1. Cytotoxicity Test

The results of the cytotoxicity assay indicated that cell viability with a CCK8 assay remained over 90% when treated with peptides at the concentrations always keeping at 0.78125 µM–400 µM, and the cell morphology was not affected. (Figure 6A). Thus, 200 µM was chosen as the highest final concentration in most of the subsequent experiments. We also used 3.125 µM, 6.25 µM, 12.5 µM, 25 µM, 50 µM, 100 µM and 200 µM of peptide 110766 in the qRT-PCR, Western blot, and IFA assay.

#### 3.4.2. Absolute Quantification Real-Time PCR

For virus pretreatment (Figure 6B,C), the virus was first incubated with different concentrations 110766 peptide (3.125 μM, 6.25 μM, 12.5 μM, 25 μM, 50 μM, 100 μM and 200 μM) for 1 h at 37 °C and then added to the cells as described in Material and Methods before processing the cells for either absolute quantification real-time PCR (qRT-PCR) assay to assess viral copy number, primer and probe sequences are listed in Table 4. Compared with the virus control, the 110766 peptide can significantly decrease viral load and block the virus into the host cell. This evidence indicates that target the region of S1-CTD of PEDV S protein can be a promising antiviral strategy.

#### 3.4.3. Western Blot and Immunofluorescence Assay

In order to further confirm the inhibitory effect of synthetic peptide 110766 on PEDV, we examined the viral protein expression using other assays, i.e., viral N protein expression by Western blot and the number of viral N-positive cells by immunofluorescence (IF) in 110766 peptide virus pretreatment condition. As shown in Figure 6D,E, both assays showed that the peptide 110766 displayed a significant inhibition on viral N protein expression through comparing with viral control. However, it is noteworthy that when the concentrations of 110766 peptides at 6.25–200 µM, the result of Western blot shows a very shallow strip, fluorescence signal is also weak under the same conditions, this shows that the results of Western blot and immunofluorescence assay are consistent and in the presence of 110766 peptide can efficiently inhibit PEDV infection.

## 4. Discussion

PEDV S protein, which plays a critical role in host affinity, virus-cell recognition, membrane fusion and entry, contains several important targets for the development of antivirals; it contains S1 and S2 subunits. Whereas the S1 subunit contains a signal peptide (SP), N-terminal domain (NTD, residues 232–471), C-terminal domain (CTD, residues 504–637), subdomains 1(SD1, residues 472–503) and subdomains 2 (SD2, residues 638–761) [23]. Related researches have shown that angiotensin-converting enzyme 2 (ACE2) [9,26] as well as dipeptidyl peptidase 4 (DPP4) [12,27,28] are the receptor for SARS-CoV and MERS-CoV, the viral receptor-binding domains (RBD) are located in the C-terminal domains of the S1 subunits(S1-CTDs) [9,12,27]. Human coronavirus 229E (HCoV-229E) [29,30] and TGEV [31,32] utilize aminopeptidase N (APN) as the host receptor and their RBDs are also located in the S1-CTD [29,32]. Therefore, the S1-CTD region in S1 subunit plays a key role in recognition of and interaction with host cell receptor. Previous studies showed that PEDV S1-CTD (residues 505–629) could interact with pAPN ectodomain (residues 63–963) [33], although pAPN was later shown not to be a functional receptor for porcine epidemic diarrhea virus infection [10,34]. However, neutralizing antibody could block viral entry into host cells by targeting RBD region in the S1 subunit from SARS-CoV-2 [35]. More importantly, S1-CTD domains are more conserved than S1-NTD domains in Variant PEDV. Therefore, PEDV S1-CTD shows a great antiviral potential for the development of peptide-based viral attachment inhibitors.

The aim of this research was to find effective affinity peptides that could block PEDV infection by make the use of molecular docking technology for virtual screening. At present, peptides are a treatment which can help us solve many problems. In terms of phage display technology, as a targeting ligand to deliver of cytotoxic drug specifically into the tumor vasculature, tumor microenvironment or into the cancer cells. On the other hand, screening of the inhibitory peptides of subtilisin by RNA display technique could maintain the stable of enzyme in liquid [36,37]. However, methods such as phage display technology, mRNA display technology, combinatorial chemistry technology, and computer-based virtual screening technology are highly dependent on high-throughput experimental screening, which readily increase workload. In contrast, the structure-based molecular docking technique of computational virtual screening possesses several important advantages, such as operation simple and fast, cuts down the workload of peptide screening, shortens the development cycle and improves the screening success rate, etc. Therefore, it is widely applied in many fields. This study represents the first attempt to make the use of molecular docking technology for virtual screening to design affinity peptides that has strong antiviral activity against PEDV infection.

CScore value, in the molecular docking, is regarded as a kind of standard to assess the binding affinities and specificity between the affinity peptides and target protein. A correlation study have also confirmed that CScore plays an important role in accurately predict the binding affinity of proteins and ligands for docking. In a word, hydrophobic interactions, electrostatic interactions, and hydrogen bonding have many interactions between the acceptor proteins and ligands as well. When analyzing the key residues of the PEDV S1-CTD protein binding site, it was found that Leu and Val at position 550, 561 play a hydrophobic role in the interaction between 110766 and PEDV S1-CTD. In addition, the hydrophobic amino acids of the peptide 110766 sequence represent 33%. These results showed that the hydrophobic interaction is not the main binding force between the peptide 110766 and the target protein.

Souzaet al. designed synthetic peptides targeting the RBD of S protein from SARS-CoV-2, and revealed that out of 650 sequences, four sequences strongly bind to RBD, changing its conformation and leading to misplaced contact with ACE2. This result suggested peptides could block the viral attachment in cells [14]. However, it has been reported that nAbs, targeting RBD and NTD of SARS-CoV-2 S protein in the S1 subunit, which its inhibitory effect is not constant. The main reason that lies in RBD and NTD region of SARS-CoV-2 S protein is more easily to be mutational frequency than other components of the S protein [38]. In other words, neutralizing antibody inhibitors designed for the RBD and NTD regions of SARS-CoV-2 S protein make it difficult to generate immunogenicity and protect against other human coronaviruses [39]. Thus, it is of importance to identify a still more conserved and key amino acid residues target in S protein for the development of human coronaviruses attachment inhibitors or nAbs.

In the current work, conserved and key amino acid residues in PEDV S1-CTD domain were chosen to design and screen affinity peptides. However, considering the purpose of our experiment was to block the adsorption of virus, the antiviral activity of polypeptide 110766 was evaluated by virus pretreatment. Results showed that peptide 110766 could significantly inhibit PEDV infection with a low concentration at 3.125 µM. The main reason for this lies in the peptide 110766 binding to the surface of infected and uninfected cells, which may restrict attachment and spread of the PED progeny viruses. On the other hand, hydrophilicity, surface accessibility, and flexibility are also the key to inhibit the effect of HR2M, HR2L, HR2P [40]. From the analysis of the lipophilic interaction between 110766 peptide and the active site of PEDV S1-CTD protein, it was found that the Arg and Lys of peptide 110766 played strong roles in the hydrophilic property as well as in the weak lipophilic property (blue). Therefore, this design of affinity peptides provides an effective strategy for PEDV control.

## 5. Conclusions

The spike protein is naturally present in trimeric structures on the surface of porcine coronavirus and is mainly responsible for adsorption and fusion while inducing host neutralizing antibodies and immune responses and is an important target for antiviral drug development. In this work, we designed, tested, and characterized five peptides targeting the PEDV S1-CTD protein using a molecular docking virtual screening technology. These peptides exhibit low cytotoxicity in vitro, and are able to inhibit PEDV attachment and entry in cells. The affinity peptide 110766, which can specifically recognize the PEDV S1-CTD protein, was successfully obtained by the molecular docking technology. This study could thus serve as a powerful tool for dissecting the attachment mechanism of PEDV and in the development of potential antiviral drugs, with broad applications for the prevention and treatment of PEDV-associated diseases.

## Figures and Tables

**Figure 1 viruses-15-00381-f001:**
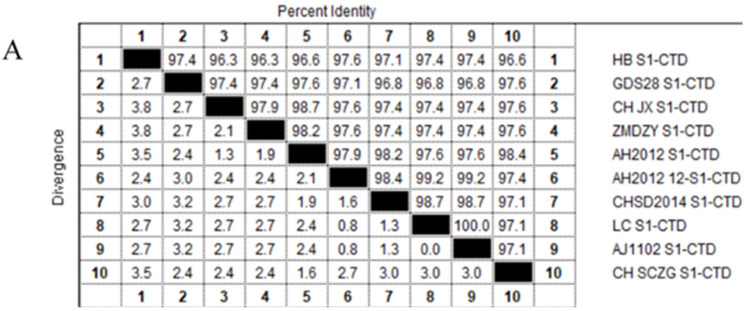
Amino acid sequence homologies from the S1-CTD regions for PEDV variant strains and overall structure of the PEDV spike protein: (**A**) Amino acid sequence homology of S1-CTD region of different PEDV mutants was higher than 96%; and (**B**) PEDV S protein was a homotrimeric class I fusion protein, describing the division of S protein structure with one of the single chains as an example. The S1 receptor-binding and S2 fusion machinery regions from PEDV is contained in spike protein. The S1 region can be divided into S1-NTD (magenta part, residues 232–471), S1-CTD (red part is in the black dotted frame, residues 504–637), this part in the 3D model of PEDV-S protein is a key target for design affinity peptides in our study, whereas green, magenta, yellow and red four parts all belong to S1 region. The S2 domain (cyan part, residues 756–1242) contains the S20 cleavage site N-terminal to the fusion peptide.

**Figure 2 viruses-15-00381-f002:**
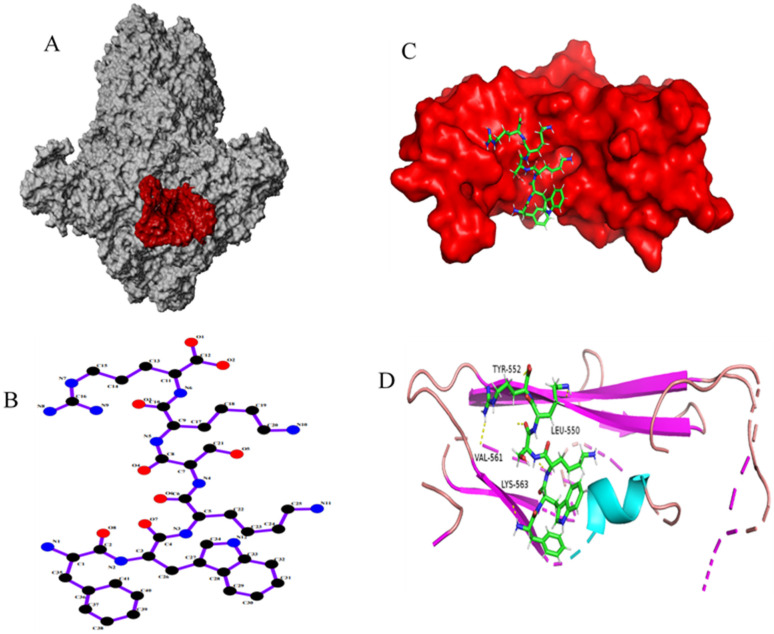
Schematic diagram of interaction between peptide and PEDV S1-CTD protein: (**A**) PEDV S protein docking active pocket display, the docking pocket of PEDV S1-CTD protein was colored red; (**B**) Interaction between peptide 110766 and PEDV S1-CTD protein; (**C**) The H-bond interactions between the ligands and target residues of protein were indicated by dotted yellow lines; and (**D**) The 2D structure of 110766-FWKSKR affinity peptide.

**Figure 3 viruses-15-00381-f003:**
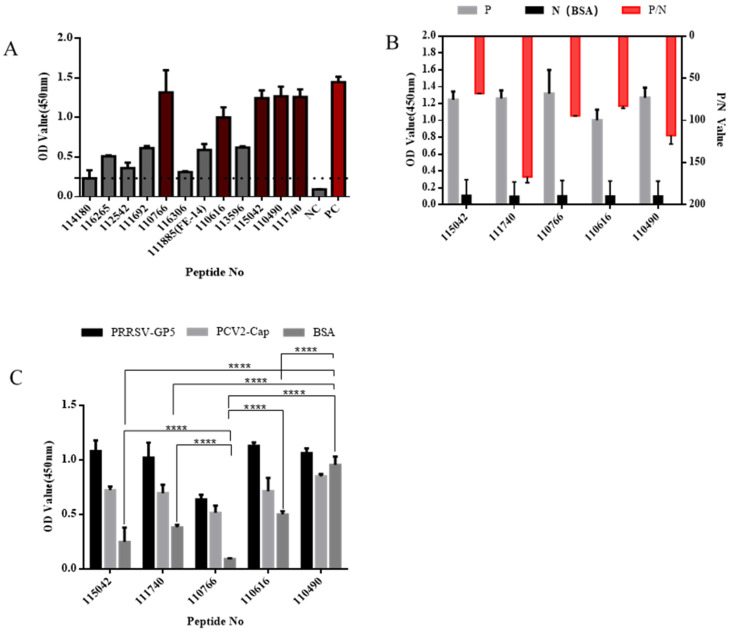
Verification of affinity and specificity of polypeptides: (**A**) The affinity of each peptide binding with the PEDV S1 protein was analyzed via ELISA. Dotted black line refers to peptide 114180 OD450 value (0.233), OD value lower than black dotted line cannot be used for subsequent experiments. The standard of criterion is based on OD450sample/OD450blank > 2.1, when the ratio is greater than 2.1, which can be determined as positive; (**B**) The P/N values of five peptides. The P/N value represents the binding specificity of the peptides (P: recombinant PEDV S1 protein; N: BSA of negative control); and (**C**) The cross-reactivity of five peptides. All the samples were tested by three parallel groups. **** *p* ≤ 0.0001.

**Figure 4 viruses-15-00381-f004:**
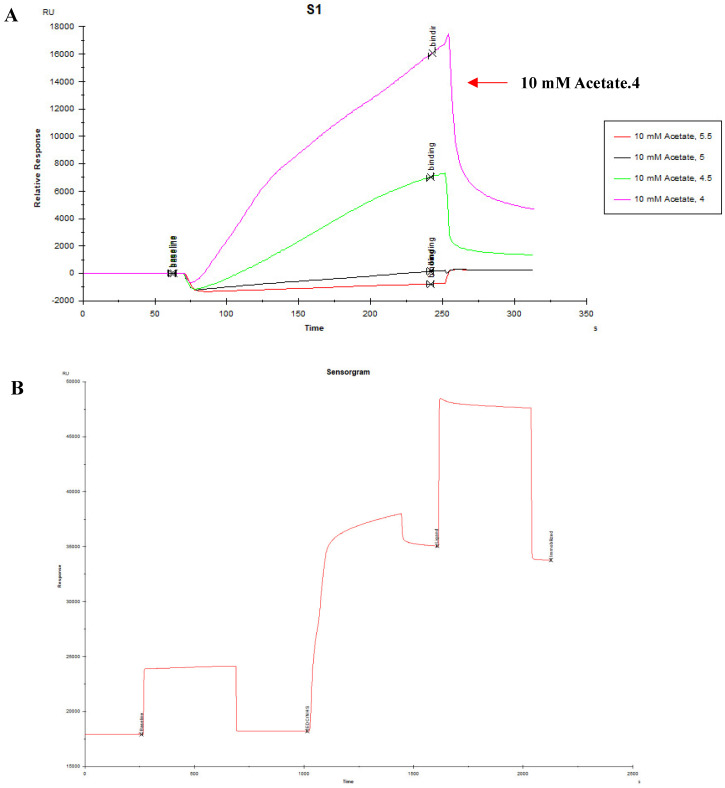
Protein-coupled CM5 chip signal. (**A**) Optimal pH signal diagram of sodium acetate buffer; (**B**) Signal diagram of protein binding CM5 chip.

**Figure 5 viruses-15-00381-f005:**
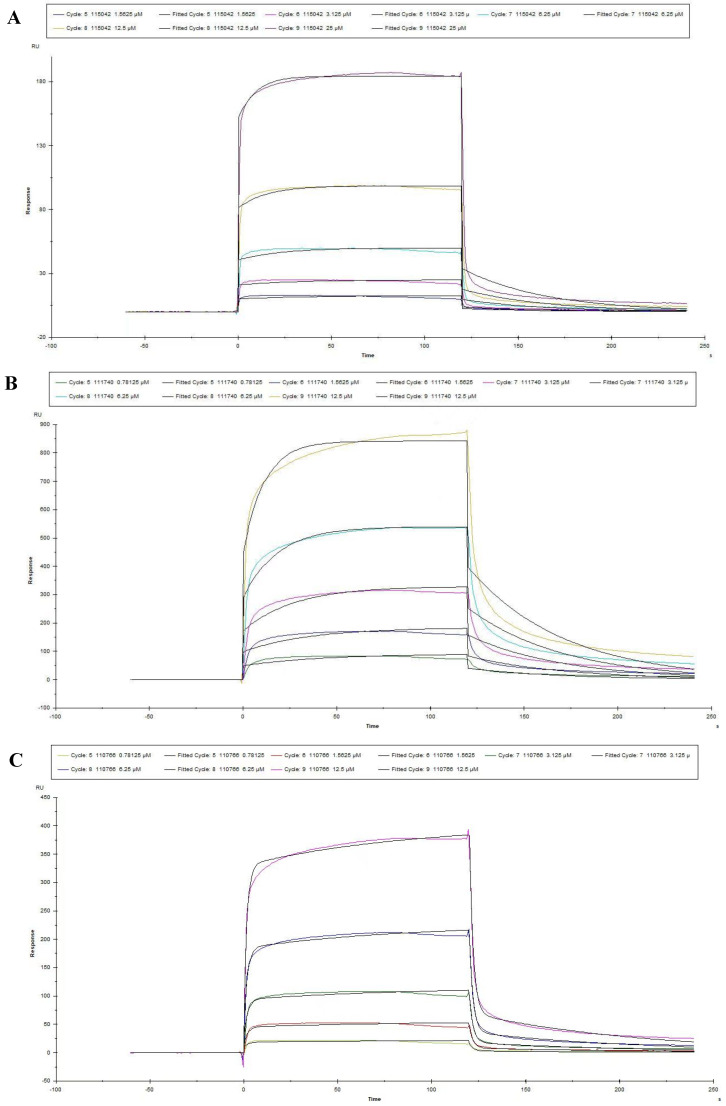
Reaction kinetic curves of five peptides with PEDV S1 protein determined by SPR assay. (**A**–**E**) represented the association and dissociation curve of 115042, 111740, 110766, 110616 and 110490, respectively. Data are fitted in a 1:1 manner by Biacore X100.

**Figure 6 viruses-15-00381-f006:**
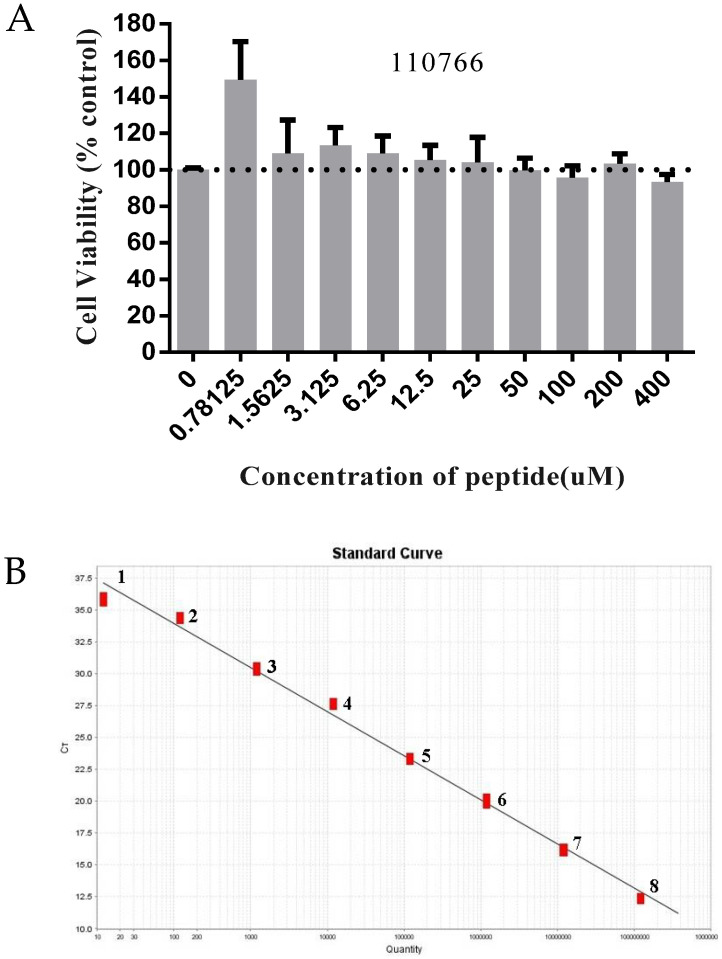
The inhibitory effect of 110766 against PEDV (CH/hubei/2016strain): (**A**) The cytotoxicity +of 110766 peptide on vero cells. Vero cell viability was measured by CCK-8. All the CCK-8 values were normalized based on the control (with no peptide 110766 which represents 100% cell viability); (**B**) Recombinant PEDV variant strain plasmid DNA standard curve. Standard Plasmid DNA were serially diluted 10-fold with distilled water, corresponding to 1.2 × 10^8^ to 1.2 × 10^1^ copies/μL, slope = −3.455, R^2^ = 0.993. 1: 1.2 × 10^1^; 2: 1.2 × 10^2^; 3: 1.2 × 10^3^; 4: 1.2 × 10^4^; 5: 1.2 × 10^5^; 6: 1.2 × 10^6^; 7: 1.2 × 10^7^; 8: 1.2 × 10^8^. (**C**) Vero cells were infected with a mixture of virus at MOI 0.01 and increasing concentrations of the peptides (3.125–200 μM), they were pre-incubated at 37 °C for 1 h, and then inoculated onto the cells for 1 h, washed 3 times/well with PBS and replaced with maintenance medium containing 2% FBS and incubated for 12 h. The positive control group (PEDV MOI = 0.01) were not treated by peptide 110766. Total RNA was then extracted from the cells, viral N gene was quantified by qRT-PCR. The results of three independent experiments are presented as the mean ± SD. The statistical analysis was performed using Graph Pad Prism 8.0. The significance was defined as *p* < 0.0001 (****); (**D**) Proteins were separated by SDS-PAGE, and Western blot membranes were probed with an antibody against either PEDV N or β-actin. M: Maker, (1–7) The concentration of 110766 peptide at 200 μM, 100 μM, 50 μM, 25 μM, 12.5 μM, 6.25 μM and 3.125 µM, respectively. 8: Virus treatment group, 9: Cell treatment group; and (**E**) Fluorescence intensity of PEDV in the presence of peptide 110766 at different concentration. (**a**) cells infected with PEDV of MOI 0.01; (**b**) mock-infected cells; (**c**) PEDV were incubated with peptide 110766 at the concentration of 200 μM, 100 μM (**d**), 50 μM (**e**), 25 μM (**f**), 12.5 μM (**g**),6.25 μM (**h**) and 3.125 µM (**i**) at 37 °C for 1 h. Scale bar, 100 µm.

**Table 1 viruses-15-00381-t001:** PEDV variant strains used in this study.

Strain Name	Countries	Time	Type	Accession
HB	China	2016	G2b	KY928065.1
GDS28	China	None	G2a	MH726372.1
CH JX	China	2015	G2a	KJ526096.1
ZMDZY	China	2011	G2a	KC196276.1
AH2012	China	2012	G2a	KC210145
AH2012-12	China	2012	G2b	KU646831.1
CHSD2014	China	2014	G2b	KX791060.1
LC	China	2012	G2b	JX489155.1
AJ1102	China	2011	G2b	JX188454.1
CH SGZG	China	2017	G2b	MH061337

**Table 2 viruses-15-00381-t002:** Sequence and CScore values of twelve candidate peptides obtained by FlexX/SYBYL virtual screening.

Peptide NO	Amino Acid Sequence	CScore	Crash	Polar
114180	FWKPKR	11.4180	−3.2727	6.5323
116265	FWKPDQ	11.6265	−3.0837	10.3922
112542	FWKHEK	11.2542	−3.1835	10.0943
111692	KHQKRC	11.1692	−3.1609	8.7458
110766	FWKSKR	11.0766	−2.8652	5.9879
116306	RKQFDK	11.6036	−2.4966	12.7206
111885	FWKYAW	11.1885	−3.5336	6.9276
110616	FWKEAK	11.0616	−2.8072	7.1044
113596	FWKCDV	11.3596	−3.8652	9.0346
115042	WHFNRP	11.5042	−3.0164	8.1665
110490	FWKQNKFLFWKQNKCL	11.0490	−4.2913	11.9966
111740	FWKHRIFWKHRI	11.1740	−3.9744	9.8984

**Table 3 viruses-15-00381-t003:** The kinetic data of affinity peptides-PEDV S1-CTD interactions in detail. A lower K_D_ value indicated a higher affinity of the peptide binding with PEDV S1-CTD.

Peptide No.	Ka (1/Ms)	Kd (1/s)	K_D_ (M)
115042	3454	0.02252	6.518 × 10^−6^
111740	5482	0.01975	3.603 × 10^−6^
110766	7.055 × 10^4^	0.01523	2.159 × 10^−7^
110616	3.050 × 10^4^	0.02355	7.721 × 10^−7^
110490	5743	0.03522	1.495 × 10^−6^

**Table 4 viruses-15-00381-t004:** Primers and probes used for the duplex TaqMan probe-based real-time RT-PCR.

Primer Name	Sequence (5′-3′)
N-F	CGCAAAGACTGAACCCACTAAC
N-R	TTGCCTCTGTTGTTACTTGGAGAT
PEDV N probe	TGTTGCCATTACCACGACTCCTGC 5′fam-3′BHQ3

## Data Availability

Not applicable.

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
