# Peer review of "Virtual Screening-Based Peptides Targeting Spike Protein to Inhibit Porcine Epidemic Diarrhea Virus (PEDV) Infection"

_viruses, 2023, doi:10.3390/v15020381_

Round 1

Reviewer 1 Report

The manuscript by Xu et al. describes the use of molecular docking technology for the virtual screening of peptides that specifically recognize the PEDV S1-CTD protein. These peptides are able to inhibit PEDV attachment and entry into cells.

The results presented in this manuscript constitute an interesting contribution to the area of vaccine and antidrug development of PEDV. Although this manuscript has information of interest, the manuscript is not clearly presented and not well organized, and there are several points that need to be clarified and revised.

Major comments:

1.     The overall structure of the manuscript is not well organized. The description of the Materials and Methods and the Results are not well related to the Figures and Tables. The authors should provide a concise and precise presentation of the experimental designs and the results, especially in sections 2.2.1 and 3.1~3.3.

2.     Section 2.2.5~2.2.7: Please describe clearly the titer of PEDV used in this study, and also the inoculation time of the Vero cells.

3.     Section 2.2.5 and line 480: The 12 h inoculation was mentioned in line 480, but not in section 2.2.5. Please check the consistency of the method.

4.     Section 2.2.5: Why used the real-time PCR assay instead of the virus titration?

5.     Figure 7C: Please describe the positive control used in this study.

6.     Figures 7D and 7E: In Western blot (Figure 7D), the PEDV N protein could be detected at the concentration of 3.125 μM of the peptide 110766. However, in the immunofluorescence assay (Figure 7E), the PEDV N protein could be detected at all concentrations of the peptide 110766 used. Please explain the inconsistency of the results.

Minor comments:

1.     Line 16: should be “porcine epidemic diarrhea virus (PEDV)”.

2.     Lines 20 and 45: should be “C-terminal domain (CTD)”.

3.     Line 52: should be “receptor-binding domain (RBD)”.

4.     Section 2: no need to separate the “Materials” and the “Methods”.

5.     Section 3.1: The descriptions of previous studies should be moved to the Discussion.

6.     Fig 1B: The alignment of amino acid sequences of variant PEDV strains was not mentioned in the Materials and Methods.

7.     Line 242: should be corrected to “12” peptides.

8.     Lines 263 and 279: should be corrected to “peptide”.

9.     Section 3.5: The Figures and Tables should be inserted into the main text close to their first citation, not in a separate section.

10.  Table 3: The format should be modified to clearly identify the primers and probe.

11.  Lines 498-499: should be in the same paragraph.

12.  Line 508: should be “RBD”.

Author Response

Reviewer #1: The results presented in this manuscript constitute an interesting contribution to the area of vaccine and antidrug development of PEDV. Although this manuscript has information of interest, the manuscript is not clearly presented and not well organized, and there are several points that need to be clarified and revised.

Major comments:

Question 1: The overall structure of the manuscript is not well organized. The description of the Materials and Methods and the Results are not well related to the Figures and Tables. The authors should provide a concise and precise presentation of the experimental designs and the results, especially in sections 2.2.1 and 3.1~3.3.

Response: The description of the materials and methods has been modified, in which Figure 3 in the original manuscript has been deleted and its elaboration was moved to the discussion section. In addition, sections 2.2.1 and 3.1-3.3 in the original manuscript were revised to make them more concise and accurate.

Question 2: Section 2.2.5~2.2.7: Please describe clearly the titer of PEDV used in this study, and also the inoculation time of the Vero cells.

Response: In the revised manuscript, section 2.2.5~2.2.7 have been changed to 2.8-2.10 section in this paper. In 2.8 section, it has described clearly the titer of PEDV used and also the inoculation time of the vero cells in this study. In 2.9-2.10 section, the mode of action of a mixture of peptide 110766 and PEDV on vero cells is the same as Section 2.8.

Question 3: Section 2.2.5 and line 480: The 12 h inoculation was mentioned in line 480, but not in section 2.2.5. Please check the consistency of the method.

Response: In the revised manuscript, section 2.2.5 has been changed to 2.8 section, “12 h of inoculation” was added to this section to maintain the consistency of the method.

Question 4: Section 2.2.5: Why used the real-time PCR assay instead of the virus titration?

Response: The purpose of the experiment is that virus copy number treated with different concentrations of peptide 110766 was detected to determine the antiviral activity of the peptide 110766 through absolute quantitative PCR, this method is also applied in some papers. In addition, the expression level of the target gene in treated with different concentrations of peptide 110766, which could also be detected by relatively quantitative PCR to determine the antiviral activity of peptide 110766. Two different detection methods, but the ultimate goal is the same.

Question 5: Figure 7C: Please describe the positive control used in this study.

Response: In the revised manuscript, figure 7C: the positive control group has been described.

Question 6: Figures 7D and 7E: In Western blot (Figure 7D), the PEDV N protein could be detected at the concentration of 3.125μM of the peptide 110766. However, in the immunofluorescence assay (Figure 7E), the PEDV N protein could be detected at all concentrations of the peptide 110766 used. Please explain the inconsistency of the results.

Response: In fact, when the concentration of peptide 110766 was 200μM, 100μM, 50μM, 25μM, 12.5μM, 6.25μM in Western blot experiment (Figure 7D), PEDV N protein could be detected, but the band was very shallow and could not be analyzed by gray scale. When the concentration of peptide 110766 was 3.125μM, PEDV N protein could be detected and the band was clear. Therefore, with the decreasing of peptide 110766 concentration, the fluorescence intensity of PEDV gradually increased in indirect immunofluorescence assay (Figure 7E). Thus the results of Western blot and indirect immunofluorescence assay were consistent.

Minor comments:

Question 1: Line 16: should be “porcine epidemic diarrhea virus (PEDV)”.

Response: In the revised manuscript, Line 16: has been modified “porcine epidemic diarrhea virus (PEDV)”.

Question 2: Lines 20 and 45: should be “C-terminal domain (CTD)”.

Response: In the revised manuscript, Line 20 and 35: has been modified C-terminal domain (CTD).

Question 3: Line 52: should be “receptor-binding domain (RBD)”.

Response: In the revised manuscript, Line 49: has been modified receptor-binding domain (RBD).

Question 4: Section 2: no need to separate the “Materials” and the “Methods”.

Response: In the revised manuscript, the format of the" Materials and Methods" in this paper has been modified as prompted by the reviewers.

Question 5: Section 3.1: The descriptions of previous studies should be moved to the Discussion.

Response: In the revised manuscript, for the descriptions as to previous studies have been moved to the discussion.

Question 6: Fig 1B: The alignment of amino acid sequences of variant PEDV strains was not mentioned in the Materials and Methods.

Response: In the revised manuscript, the alignment of amino acid sequences of variant PEDV strains has added in the Materials and Methods.

Question 7: Line 242: should be corrected to “12” peptides.

Response: In the revised manuscript, Line 215 has been corrected to "12" peptides.

Question 8: Lines 263 and 279: should be corrected to “peptide”.

Response: In the original manuscript, five peptides in line 263 (located in line 236 of the revised manuscript) should not be changed to five peptide, five peptides represent the plural. 110766 peptides in line 279 (located in line 281-282 of the revised manuscript) has been changed to 110766 peptide.

Question 9: Section 3.5: The Figures and Tables should be inserted into the main text close to their first citation, not in a separate section.

Response: In the revised manuscript, the figures and tables have been inserted into the main text close to their first citation.

Question 10: Table 3: The format should be modified to clearly identify the primers and probe.

Response: In the revised manuscript, the format of primer and probe in table 4 has been modified.

Question 11: Lines 498-499: should be in the same paragraph.

Response: In the revised manuscript, line 498 and line 499 have been merged (located in line 363 of the revised manuscript).

Question 12: Line 508: should be “RBD”.

Response: In the revised manuscript, Line 382: has been modified RBD.

Reviewer 2 Report

In this study, authors designed and synthesized the peptide targeting porcine epidemic diarrhea virus (PEDV)S1-CTD region by making use of computer virtual screening technology, and screened the optimal peptide via Enzyme Linked Immunosorbent Assay (ELISA) and Surface Plasmon Resonance (SPR) experiment. This method is relatively novel for the study of the relationship between virus and host. But there are also the following problems.

1. There should be a space between all numbers and unit symbols in this paper. Please check carefully and correct them.

2. In figure 4A, please explain what is the dotted line at OD value 0.5.

3. In figure 4C, the marked of significant differences is confused and a positive control group and viruses of the same genus as PEDV should be added instead of PRRSV and PCV.

4. In reference part, there are not Journal names, years and so on in reference 5, 12, 13, 22, 23, 25, 26, 29, 31. there is not issue volume and page number in reference 39.

Author Response

Question 1: There should be a space between all numbers and unit symbols in this paper. Please check carefully and correct them.

Response: In the revised manuscript,a space exists between all numbers and unit symbols.

Question 2: In figure 4A, please explain what is the dotted line at OD value 0.5.

Response: In the revised manuscript, figure 4A is modified and the meaning of the black dotted line is explained in note.

Question 3: In figure 4C, the marked of significant differences is confused and a positive control group and viruses of the same genus as PEDV should be added instead of PRRSV and PCV.

Response: In the revised manuscript, figure 4C has been re-modified. On the one hand, transmissible gastroenteritis of pig, porcine delta coronavirus and porcine epidemic diarrhea virus all belong to coronavirus, but the homology from S protein S1 region is relatively low. And considering the possible of cross-reaction with peptides targeting the PEDV S1-CTD region. Eventually, the GP5 protein of porcine reproductive and respiratory syndrome virus, porcine circovirus Cap protein and bovine serum albumin were selected for the experiment. On the other hand, Figure 4A shows the part peptides has higher affinity with PEDV S1 protein, especially 110766. Thus this group is not added as a positive control in Fig.4C.

Question 4: In reference part, there are not Journal names, years and so on in reference 5, 12, 13, 22, 23, 25, 26, 29, 31. there is not issue volume and page number in reference 39.

Response: In the revised manuscript, the format of references has been revised.

Reviewer 3 Report

The paper presents a method of make the use of molecular docking technology for virtual screening of affinity peptides that specifically recognize the target antigen protein. It is a topic of interest to the antiviral drug development researchers and other related fieldsbut the paper needs very significant improvement before acceptance for publication. My detailed comments are as follows:

1.       Monoclonal antibodies against S1 protein of porcine epidemic diarrhea virus was mentioned in materials and methods, however, 3.3 result part showed that porcine epidemic diarrhea virus S protein was regarded as positive control in indirect ELISA test, pre and post statement is nonuniform.

2.       The title of the paper is that virtual screening-based peptides targeting spike protein to “inhibit porcine epidemic diarrhea virus (PEDV) infection”. But in the introduction part, five peptides are called 115042, 111740, 110766, 110616 and 110490 that were able to “inhibit PEDV attachment”, pre and post statement is nonuniform.

3.       Writing formats of pH and affinity (KD) in the paper exist errors, please correct such errors in the text one by one.

4.       The paper needs to be improved the grammar. The main problem is that part statement is not smooth, especially the material and method in section 2 should be re-organized. Also consider to avoid the repeated emergence of using words, the same problem should be avoided in the paper.

5.       This paper listed many references that are mainly related to coronavirus, but the format of a few references needs to be further improved, such as reference 10: Ojkic, D.; Hazlett, M.; Fairles, J.; Marom, A.; Slavic, D.; Maxie, G.; Alexandersen, S.; Pasick, J.; Alsop, J.; Burlatschenko, S., The first case of porcine epidemic diarrhea in Canada. (0008-5286 (Print))”.

6.   discussion part need to imporve.

Author Response

Reviewer #3: The paper presents a method of make the use of molecular docking technology for virtual screening of affinity peptides that specifically recognize the target antigen protein. It is a topic of interest to the antiviral drug development researchers and other related fields,but the paper needs very significant improvement before acceptance for publication. My detailed comments are as follows.

Question 1: Monoclonal antibodies against S1 protein of porcine epidemic diarrhea virus was mentioned in materials and methods, however, 3.3 result part showed that porcine epidemic diarrhea virus S protein monoclonal antibodies was regarded as positive control in indirect ELISA test, pre and post statement is nonuniform.

Response: In the revised manuscript, the result of section 3.2: S protein monoclonal antibodies has been modified to S1 protein monoclonal antibodies.

Question 2: The title of the paper is that virtual screening-based peptides targeting spike protein to “inhibit porcine epidemic diarrhea virus (PEDV) infection”. But in the introduction part, five peptides are called 115042, 111740, 110766, 110616 and 110490 that were able to “inhibit PEDV attachment”, pre and post statement is nonuniform.

Response: In the revised manuscript, in the introduction part “inhibit PEDV attachment” has been modified to inhibit PEDV infection”.

Question 3: Writing formats of pH and affinity (KD) in the paper exist errors, please correct such errors in the text one by one.

Response: In the revised manuscript, the writing format of pH and affinity (KD) has been modified in the paper.

Question 4: The paper needs to be improved the grammar. The main problem is that part statement is not smooth, especially the material and method in section 2 should be re-organized. Also consider to avoid the repeated emergence of using words, the same problem should be avoided in the paper.

Response: In the revised manuscript, related issues in the material and method part have been revised.

Question 5: This paper listed many references that are mainly related to coronavirus, but the format of a few references needs to be further improved, such as reference 10: “Ojkic, D.; Hazlett, M.; Fairles, J.; Marom, A.; Slavic, D.; Maxie, G.; Alexandersen, S.; Pasick, J.; Alsop, J.; Burlatschenko, S., The first case of porcine epidemic diarrhea in Canada. (0008-5286 (Print))”.

Response: In the revised manuscript, the format of references has been revised.

Question 6: Discussion part need to imporve.

Response: In the revised manuscript, discussion part has been imporved.

Round 2

Reviewer 3 Report

Paper can be accepted